# Association between pregnancy planning or intention and early child development: A systematic scoping review

Jorge Andrés Delgado-Ron [1,2]*, Magdalena Janus [1,3]

1 School of Population and Public Health, Faculty of Medicine, University of British Columbia, Vancouver, British Columbia, Canada, 2 Faculty of Health Sciences, Simon Fraser University, Vancouver, British Columbia, Canada, 3 Offord Centre for Child Studies, Department of Psychiatry and Behavioural Neurosciences, McMaster University, Hamilton, Ontario, Canada

* andres@delgado.ec

**Data Availability Statement:** All relevant data are within the paper and its Supporting Information files.

## Abstract

The Sustainable Development Goals have a specific target to "ensure that all girls and boys have access to quality early childhood development" by 2030. There is sparse literature regarding the impact of pregnancy intention (wantedness and timing) or planning on child development. We aimed to summarize the evidence measuring the association between unintended or unplanned pregnancy and child development for children aged 3 to 5. We included studies that compared developmental outcomes of children from unwanted, unplanned, or mistimed pregnancies to those of children from wanted or planned pregnancies. In April 2022, we searched Ovid MEDLINE (from 1946), EMBASE (from 1974), and SCOPUS (from inception) without language restrictions. One reviewer tabulated data on country and year of study, population, sample and sampling method, age of participants, exposure, date of exposure retrieval, developmental outcome(s), assessment (direct or through third parties), statistical methods (including covariate selection methods), and effect estimates into bespoken data tables. Our scoping review identified 12 published studies reporting on 8 "cohorts" (range: 1963–2016) with information on approximately 39,000 children born mostly in developed countries. Overall, unwanted/unplanned pregnancies seemed to be associated with poorer child development when compared with wanted/planned pregnancies. Mistimed or delayed pregnancies correlated with weaker effects in the same direction. We concluded that achieving the target for SDG 4, related to child development, might entail policies that create environments supportive of wanted conception and access to safe abortion.

## Introduction

When a Lancet report uncovered that one in three children under five from low-and-middle-income countries were at risk of not reaching their full potential due to the risk of poor development [1], researchers and policy actors mobilized across the globe [2] to ensure that children were prioritized within the new Sustainable Development Goals. As a result, under SDG 4 (focused on education), there is now a specific target to "ensure that all girls and boys have

**Funding:** The authors received no specific funding for this work.

**Competing interests:** The authors have declared that no competing interests exist.

access to quality early childhood development" by 2030 [3, 4], given the life-long consequences of early child development (ECD) for both mental health and physical well-being. However, the COVID-19 pandemic has undone much of the progress made in recent decades by restricting access to healthcare, facilitating a loss of caregivers, and placing most of the economic burden on fragile populations [5]. Children are expected to suffer consequences irrespective of whether they get infected [6]. Such challenges are particularly pronounced in homes already facing difficulties, potentially impacting maternal perspectives on present or future pregnancies, which can range from wishes to delay conception and temporary pregnancy regrets to long-lasting perceptions of pregnancy and birth as an undesired event [7, 8].

While experts have warned about the potential of stressful environments on child development [6], there is sparse literature regarding the impact of pregnancy intention (wantedness and timing) or planning on child development. The first review on the topic was published in 1986 and documented findings from Prague and Northern Finland. This early work reported on the effects of wantedness on children at ages 9 and 14, respectively [9]. Almost a decade later, a new review commissioned by the US National Institute of Medicine's Committee on Unintended Pregnancy [10] was published. Apart from describing the outcomes of "most extreme examples of unwanted conceptions" (i.e., children whose mothers were denied abortions), the commission referenced the preliminary findings of the first study exploring the outcomes of children at ages 2 and 5 [11]. A more recent review by Šulová and Fait [12] also relied heavily on literature from a single study of a subgroup of children whose mothers had unintended pregnancies after being denied abortions and failed to include several studies that had been published at the time. Neither of these reviews reported a systematic search, and their findings are considerably dated.

The overturn of *Roe v. Wade* by the United States Supreme Court has reignited the debate regarding abortion rights. Women experiencing miscarriages are already struggling to access appropriate healthcare as a consequence [13], and those seeking abortions for unwanted pregnancies are likely to also face criminalization and prosecution under charges of "homicide, feticide, assault, and child abuse laws" in at least 18 states [14]. Denial of abortion services increases anxiety and stress and lowers self-esteem among women who seek this service, often leading to chronic distress years after the event [15]. These women also face a four-fold increase in the odds of becoming poor [16] and often fail to suppress cigarettes or drug consumption during an unwanted pregnancy [17]. Such effects are relevant to a woman's health and may influence children born from pregnancies that were initially perceived as unwanted. Although past research has shown that feelings about past pregnancies can change over time and after pregnancy, population-level estimates remain consistent and retrospective reporting is considered a valid tool to assess pregnancy intention [8, 18].

Given the importance of early child development, the intertwined nature of family planning and women's rights, the impacts of the COVID-19 pandemic (known and not yet known) and the current tensions in fragile political systems (e.g., *Roe vs. Wade*), more research is required on the association between the pregnancy intention—often associated with social, economic, and familial contexts—and developmental outcomes at early ages. Our systematized review aimed to summarize the literature measuring the association between unintended or unplanned pregnancy and child development for children aged 36 to 59 months, including effect estimates and analytical methods.

## Methods

We followed the Preferred Reporting Items for Systematic Reviews and Meta-Analyses extension for Scoping Reviews Checklist (PRISMA-ScR) to report this review [19]. The protocol was reviewed and approved by the study team but not published on the web.

### Eligibility criteria

We included observational studies that reported developmental outcomes about children 36 to 59 months old whose mothers had an unwanted, unplanned, or mistimed pregnancy. We primarily aimed to include studies that compared these children to controls of the same age born from planned/wanted pregnancies. We excluded studies that reported exclusively on physical health (e.g., height, weight, or malnutrition). However, composite outcomes that include physical health as part of their assessment were included. We also excluded articles whose primary intent was to predict child development without a causal framework to guide the analysis.

### Search strategy and information sources

Our search strategy used natural and controlled language to identify a sample of representative studies. For the former, we identified a set of ten references of population-level studies through a naive search and *a priori* knowledge. We then used the '*Litsearchr*' package [20] to "extract" search terms from the articles' title, abstract, and associated metadata through text analysis. Our specialized controlled vocabulary corresponded to the Medical Subject Headings terms from the National Library of Medicine for our inclusion criteria. Our final strategy (S1 Appendix) had a high sensitivity (90%) compared to the original sample. We carried out the searches without language restrictions on April 19, 2022, in Ovid MEDLINE (from 1946), EMBASE (from 1974), and SCOPUS (from inception). Finally, we conducted one-step backward- and forward-reference searching of relevant studies.

### Study selection and data extraction

Upon de-duplicating our references, we imported our search results into Rayyan, an online tool that supports systematic reviews [21]. We screened the title and abstract for eligibility. At this point, age was not considered as an exclusion criterion because we assumed some articles might report interim results at a younger age. If the article appeared to meet all other criteria, we sought the full text and screened for eligibility screening for studies in the corresponding age bracket only.

For articles in a language other than English, a member of the reviewer team searched for translated versions of the study, used automated translation via Google's Neural Machine Translation, or sought assistance from a native speaker in reading and interpreting the result, depending on availability.

### Data items

We extracted the following data items: country and year of study, population, sample and sampling method, age of participants, exposure, date of exposure retrieval, developmental outcome(s), assessment (direct or through third parties), statistical methods (including covariate selection methods), subgroup analyses, and effect estimates. One reviewer tabulated the data into bespoken tables with fields agreed upon by the review team.

### Effect measures and synthesis

We classified the developmental outcome(s) according to each study's closest Early Child Development Index domain(s) [22] following the findings from a recent concurrent validity study [23]. In the absence of formal criteria, the reviewers used their judgment to place the outcome measure within the categories mentioned above.

We reported differences in estimate, Chi-Square tests, Rate Ratio, Odds Ratio (OR), and Relative Risk (RR) plus 95% confidence intervals if available. When authors reported no

difference without summary statistics, we assumed a *P*-value above 0.05, and no data conversion or imputation was performed. For nested regression models, we reported those adjusted for socioeconomic variables but not for child characteristics or parenting style. We did not perform a meta-analysis, given the heterogeneity of sampling techniques, assumptions, and statistical methods. Instead, we provide a narrative synthesis. In addition, no formal risk of bias assessment was carried out in this review.

## Results

### Study selection

Our search yielded a total of 215 records; an additional record was identified through citation review. All but two records written in a language other than English (4 Czech, 3 German, 3 Spanish, 1 French, 1 Polish, 1 Chinese, 1 Bulgarian) provided either an English abstract or had published an English version of the article and were considered duplicates. After removing duplicates, we identified 153 records whose titles and abstracts were subsequently screened. We assessed the full text of 48 records (including two articles in German), of which 36 were excluded. Most records were excluded because they were conducted in children aged six or older (S2 Appendix). The final sample of 12 records represents eight different samples or "studies" (Fig 1).

### Study characteristics

**Population, study design, and sampling.**   The included records spanned from 1976 to 2021 and included approximately forty thousand children from six countries: New Zealand (NZ), Germany, Canada, Brazil, the United States, and the United Kingdom.

Two prospective studies recruited birth cohorts [24, 25], one in Christchurch, NZ, and another through all twenty-one university clinics across Germany. Two others were retrospective and used a cross-sectional design, sampling from the lists of the Wellington-Dufferin-Guelph Health Unit in Ontario, Canada [26] and the public preschool system in Embu das Artes in Brazil [27].

All other studies used data from nationally representative surveys of children—the Maternal and Infant Health Survey [28, 29] and the Early Childhood Longitudinal Study [30] from the United States and the Millennium Cohort Study [31–33] from the United Kingdom—or young adults (aged 14 to 22 during the first wave): The Longitudinal Survey of Youth [11, 34].

**Assessment of developmental outcomes.**   The first use of a valid instrument of early child development (ECD) in the included studies occurred in NZ; Fergusson and Horwood [25] used scales correlated with the Stanford-Binet Intelligence Scale, a tool that measures cognitive ability and intelligence. All subsequent studies have used validated instruments (See Table 1), measuring at least two domains of ECD. Most studies relied on parental reports or parental diaries. The only assessment of ECD that did not rely on parental reports was conducted during the Longitudinal Survey of Youth, reported by Baydar [11] and Joyce and colleagues [34]. The first study evaluating the effect of unwanted pregnancy on all domains of ECD occurred in Brazil in 2016 [27].

**Pregnancy intention/planning.**   Most studies (75%) assessed the effect of pregnancy intendedness, while the remainder reported on the effects of planning. Mothers were asked about their pregnancy at various times, ranging from pregnancy to five years after birth (average: 1.75 years). If we consider the trends over time of nationally representative studies, the prevalence of *unwanted* pregnancy seems to have increased over time from 5% [11] to 10% [34] in the same USA cohort and to 13.8% in a different cohort and using a different sampling method (children instead of mothers) [30]. Conversely, the prevalence of *unplanned*

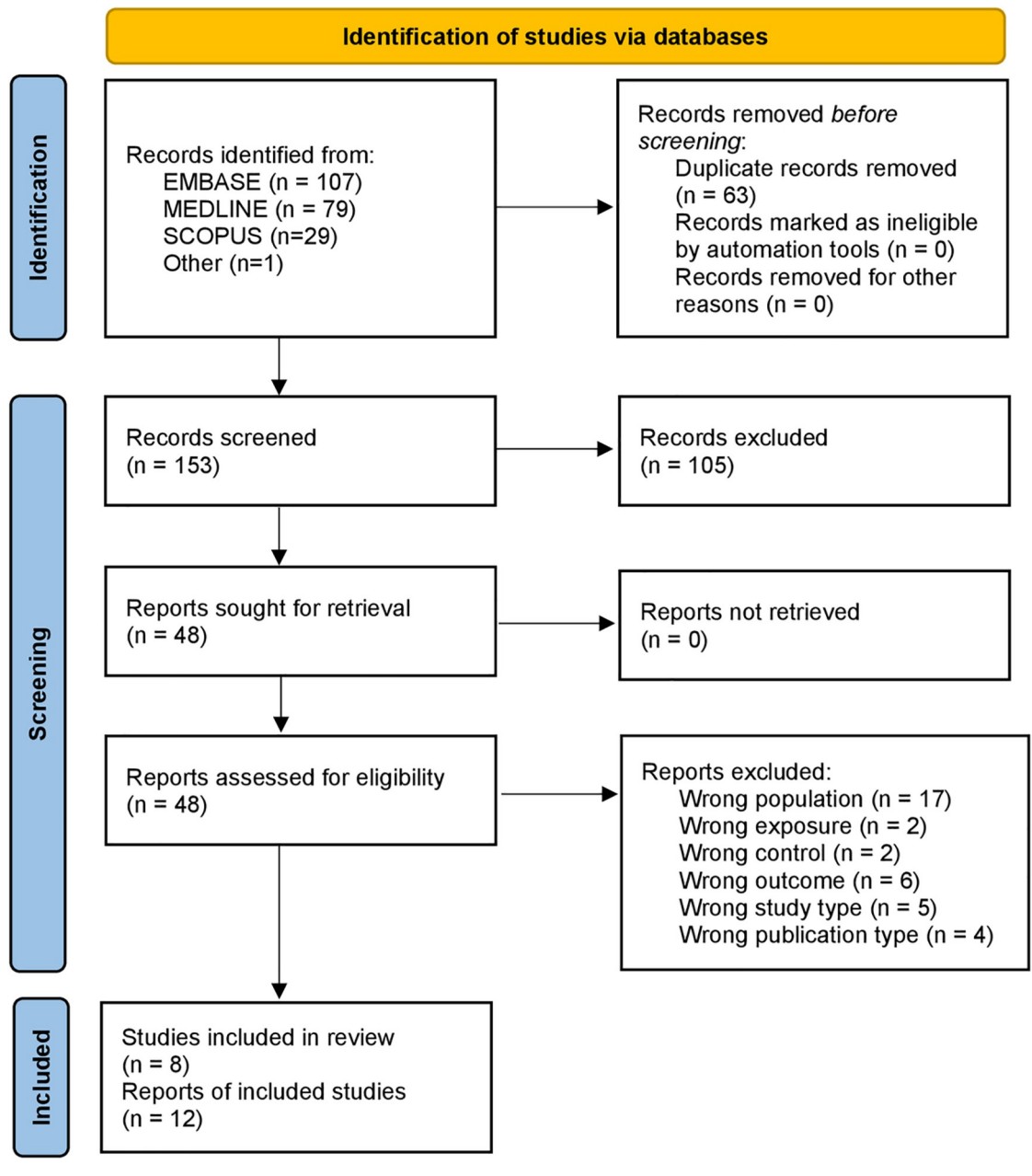

**Fig 1. PRISMA 2020 flow diagram for the included studies.**

pregnancy was higher in an older study [25] compared to more recent reports [31–33]. Apart from those, most population studies report a relatively high prevalence of *mistimed* pregnancies—wanting to be pregnant "later" or at a different time—ranging from 26% to 46.3%. A few studies supplemented their analysis with measures of contraceptive use or other forms of birth control [24, 25, 29, 30], marital status [26], ideal family size [34], use of fertility assistance [31, 33], and paternal intendedness (or agreement) [25, 30].

 **Estimation methods.** Studies of children born in 1976 or earlier lacked advanced statistical methods, relying on chi-square tests [24] or unadjusted risk ratios [26]. These earlier assessments assumed marital status, the previous number of pregnancies, number of siblings, and

**Table 1. Matrix of covariates and regression type of studies using multivariate regression or regression-like methods.**

| Group | First author | Ferguson et al. | Baydar | Joyce et al. (A) | Joyce et al. (B) | Hummer et al. (2) | Hummer et al. (3) | Hummer et al. (4) | Crissey | Carson et al. (1A) | Carson et al. (2A) | Carson et al. (3A) | Carson et al. (4A) | Carson et al. (2013) | Rochebrochard (2) | Rochebrochard (3) | Rochebrochard (4) | Saleem et al. | Jang et al. |
|---|---|---|---|---|---|---|---|---|---|---|---|---|---|---|---|---|---|---|---|
| **Covariate** | | | | | | | | | | | | | | | | | | | |
| **Mother** | Age | X | X | | | | | X | X | X | X | X | X | X | X | X | X | X | X |
| | Marital | X | X | | X | | | X | X | | X | X | X | X | X | X | X | | |
| | Occupation/Class | X | X | | X | | | | | X | X | X | X | X | X | X | X | X | X |
| | Education | X | X | X | X | | | X | X | | | | | X | X | X | X | X | |
| | Ethnicity | X | X | X | X | | | | | | | | | | | | | X | X |
| | Intelligence | | X | | X | | | | | X | X | X | | | | | | | |
| | Depression | | X | | X | | | | | | X | X | X | | | | X | | X |
| | Health | | | | | | | | | | X | X | X | | | | | | |
| | Attachment | | | | | | | | | | | | X | | | | X | | |
| | Infancy | | | X | X | | | | | | | | | | | | | | |
| | Postpartum Depression | | | | | | | | | | | X | X | | | | | | |
| | Involvement | | | | | | | | | | | X | X | | | | X | | |
| | Religiosity | | | | X | | | | | | | | | | | | | | |
| **Child** | Sex at birth | X | X | X | X | X | X | X | X | X | X | X | X | X | | X | X | | X |
| | Birth order | X | X | | X | X | X | X | X | X | X | X | X | X | X | X | X | | |
| | Ethnicity/Language | | | | | X | X | X | X | X | X | X | X | | X | X | X | | |
| | Birth weight/premature | X | X | | | | X | X | X | | | X | X | | | X | X | | |
| | Risky pregnancy (†) | X | X | | | X | X | X | | | X | X | X | | | X | X | | |
| | Age (‡) | | | | | | | | | | | | | X | | | | | |
| | Breastfed | | | | | | | | | | | X | X | | | | X | | |
| | Childcare | | | | | | | | | | | | X | | | | | | X |
| **Father** | Age | | | | | | | | | | X | X | X | | | | | | |
| | Education | | | | | | | | | | X | X | X | | | | | | |
| | Depression | | | | | | | | | | | | X | | | | | | |
| | Involvement | | | | | | | | | | | | X | | | | | | |
| **Household** | Income | X | X | | | | | X | X | X | X | X | X | | X | X | X | X | |
| | Region | | | X | X | | | | | | | | | | | | | | |
| | Rural | | | X | X | | | | | | | | | | | | | | |
| | Clustering | | | X | X | | | | | | | | | | | | | | |
| **Type of regressions of the included studies** | | | | | | | | | | | | | | | | | | | |
| | Linear (¥) | Y | Y | Y | Y | | | | | Y | Y | Y | Y | Y | | | | | |
| | Logistic | | | | | | | Y | Y | | | | | Y | Y | Y | Y | Y | |
| | Multinomial | | | | | Y | Y | | | | | | | | | | | | Y |
| | Poisson | | | | | | | | | | | | | | | | | | Y |

† Includes risk of birth defects and consumption of harmful substances during pregnancy.

‡ Some outcome scales do not require age adjustment.

¥ Includes multiple classification analyses.

housing quality as potential confounders between unwanted pregnancies and developmental outcomes. Therefore, summary statistics were produced for some subgroups with selective reporting for both studies.

Fergusson and Horwood [25] used multiple classification analysis, a multivariate technique similar to linear regression that relaxes the linearity assumption but allows adjustment for potential confounders. All subsequent studies used regression models (Table 1). Covariates controlled by at least half of the 10 studies using multivariate regression or regression-like methods were maternal age (n = 10), marital status (n = 7), socioeconomic position (n = 7), ethnicity (n = 5), and depression (n = 5), the child's sex (n = 8), birth order (n = 7), and whether it had low weight at birth or was premature (n = 5), and household income (n = 7). Notably, paternal characteristics were ignored by 9 of the 10 studies using regression-like methods.

All studies using survey data provided weighted estimates that accounted for sampling procedures. However, sampling techniques and sample size undoubtedly influenced the overall estimates. Jang and colleagues [27], for example, did not use a nationally representative sample. Instead, they relied on a registry that excluded children who did not attend preschools regularly. If unwanted children were more likely to belong to this group, it would likely bias this study's results toward the null. Similarly, differences in exposure and outcome reports from studies sampling mothers instead of children and studies recruiting at birth instead of those who do it later in life are likely to produce different estimates.

Some studies sometimes offered more than one estimate. Joyce and colleagues [34], for instance, examined both within-mother variation (i.e. sibling differences) and "between-family" variations where average outcomes for all siblings in a family were regressed on the proportion of children from unintended pregnancies. They also used a nested model to examine the effect of certain covariates on their estimands, a practice that was later followed by others [28, 31, 32].

The Early Childhood Longitudinal Study also measured the effect of parental disagreement regarding planning status on child development [30]. Jang and colleagues [27] used a doubly robust estimation method with inverse probability weighting, incorporating socioeconomic status, mother's history of emotional problems, mother's race, and mother's age at childbirth as determinants of pregnancy intention. This method reduces the probability of errors due to misspecification of the exposure or outcome model, but not both.

## Effect direction and estimation

None of the included studies reported advantageous outcomes for children born from unintended or unplanned pregnancies. The studies reported either no differences between groups or increased odds of inadequate development for all domains but 'approaches to learning' assessed by one study only. This is true whether one considers each exposure individually or together. The socio-emotional and literacy-numeracy domains had been studied separately more often. In turn, 'physical' and 'approaches to learning' had only been studied along other domains. The former showed poorer development in two out of three reports where it was assessed (Table 2).

Several studies found a meaningful association between *unwanted* pregnancies and inadequate development, using the Preschool and Kindergarten Behavior Scale [30], the Denver Developmental Scale [28, 29], and the Behaviour Problems Index [34]. While the Peabody tests for vocabulary, math and reading produced mixed results, these are likely attributable to differences in sampling, given that Baydar [11] used a subsample of Joyce and colleagues' [34] participants. No association was found using the Engle Scale [27].

**Table 2. Summary of included studies and results*.**

| Cohort, pregnancy year(s), and country | Participants for the reported outcome | Exposure (%); timing | Outcome assessor | Literacy-numeracy domain results | Physical domain results | Social-emotional domain results | Approaches to learning domain results |
|---|---|---|---|---|---|---|---|
| German Research Foundation, 1963–1976, Germany [24]. | 7,293 children evaluated at 2, 9, 18, and 36 months | Unwanted (28.8%); during pregnancy | Mother | — | — | Prolonged thumb sucking:<br>N: ↔<br>($p > .05$)<br>Destroying toys:<br>N: ↔<br>($p > .05$) | — |
| Wellington County, Ontario, 1964, Canada [26]. | 1,300 Children aged 4 to 5 | Unwanted (19.5%); ~4.5 years | Mother | Developmental milestones:<br>N: ↔/↑<br>uRR: 1.27 | | Overreacting behaviour disturbance:<br>N: ↔/↑<br>uRR: 1.27 | — |
| Christchurch Child Development Study, 1977, New Zealand [25]. | 1,106 children evaluated at 0, 4, 12, 24, and 36 months | Unplanned (36%); at birth | Parents | Experiences Scale:<br>NP: ↔<br>β: 0.38 ($p > .05$)<br>Activities Scale: NP: ↔<br>β: 0.05 ($p > .05$) | — | — | — |
| National Longitudinal Survey of Youth, 1979 (USA) | 930 children aged 4 [11] & 5,329 children aged 3 to 11 [34][a] | Mistimed (35%) or unwanted (5%); ~85% during first two years | Interviewer; Mother | *Peabody vocabulary test* [11]:<br>N: ↔<br>β: 0.32 ($p > .05$)<br>M: ↔<br>β: -0.48 ($p > .05$)<br>Peabody tests:<br>N: ↔/↑<br>β: -0.41 (-1.02, 0.19)<br>M: ↔<br>β: -0.18 (-0.6, 0.24)<br>math<br>N: ↔/↑<br>β: -0.57 (-1.18,0.02)<br>M: ↔<br>β: -0.02 (-0.48,0.43)<br>*reading*<br>N: ↑<br>β: -1.62 (-2.98, -0.25)<br>M: ↑<br>β: -0.88 (-1.78, 0.02)<br>*vocabulary* | — | Behaviour Problems Index:<br>N: ↑<br>β: -0.83 (-1.53, -0.13)<br>M: ↑<br>β: -0.66 (-1.05, -0.27) | — |
| National Maternal and Infant Health Survey, 1988, USA [28, 29]. | 6,640 children with a mean age of 34.9 months | Delayed (25%), mistimed (36.1%), unwanted (6.7%), or unintended (57.3%); 16 to 18 months. | Mother | Denver developmental scale [29]:<br>N: ↔/↑<br>OR: 1.13<br>M: ↔/↑<br>OR: 1.04<br>[28]:[b]<br>D: ↔<br>OR: 1.07<br>M: ↑<br>OR: 1.36 ($p \leq .05$)<br>N: ↑<br>OR: 1.89 ($p \leq .05$)<br>U: ↔<br>1.06 | | | — |

*(Continued)*

**Table 2.** (Continued)

| Cohort, pregnancy year(s), and country | Participants for the reported outcome | Exposure (%); timing | Outcome assessor | Literacy-numeracy domain results | Physical domain results | Social-emotional domain results | Approaches to learning domain results |
|---|---|---|---|---|---|---|---|
| Millennium Cohort Study, 2000 (UK) | 12,136 children interviewed at ages 3 and 5 [31–33] | Mistimed (26%) or unplanned (15%); 9 months | Main parent; second co-resident parent; older siblings (England only) | Bracken Scale: *NP*: ↑ OR: 1.15 (0.99;1.34) British Ability Scales II: NP: ↑ β: −5.3 (−6.4, −4.3) M: ↔ β: −2.8 (−3.7, 2.0) Verbal (age 3) NP: ↑ β: −4.9 (−5.8, −4.0) M: ↑ β: −2.9 (−3.7, -2.2) Verbal (age 5) NP: ↑ β: −2.0 (−2.8, −1.2) M: ↑ β: −1.8 (−2.4, -1.2) Non-verbal (age 5) NP: ↑ β: −4.7 (−6.0, −3.3) M: ↑ β: −2.9 (−3.9, -2.0) Spatial (age 5) | — | At-risk according to the Strengths and Difficulties Questionnaire: *NP (age 5)*: ↑ 1.55 (1.28, 1.89) *M (age 5)*: ↑ OR: 1.30 (1.09, 1.56) | — |
| Early Childhood Longitudinal Study, 2001 (USA) | 4,000 children aged 5 to 6 [c] [30] | Mistimed (46.3%) or unwanted (13.8%); 9 months | Teacher | — | — | Preschool and Kindergarten Behavior Scales: *N*: ↑ β: -0.16 (-0.28, -0.04) *M*: ↔ β: -0.03 (-0.11, 0.04) | — |
| Embú das Artes Preschool Mental Health Study, 2016 (BR) | 1,034 children aged 4 and 5 [27] | Mistimed (46.0%) or unwanted (13.4%); 4 to 5 years | Mother | Engle Scale: *U*: ↔ aRR: 1.00 (0.93–1.07) *N*: ↔ aRR: 1.02 (0.88–1.18) *M*: ↔ aRR: 0.96 (0.86–1.08) | | | |

USA: United States of America; UK: United Kingdom; BR: Brazil. T: time of report after birth.

* We present the outcome result of each study according to the closest domain from the Early Child Development Index [22], adjusting for socioeconomic variables (when available) but not for child characteristics or parenting style. Arrows represent an increased risk of poor development. D: Delayed; U: Unintended; N: Unwanted; M: Mistimed; NP: Unplanned. OR: Odds Ratio; uRR: Unadjusted Relative Risk; aRR: Adjusted Relative Risk; β: Effect size.

a. The earliest valid outcome for all children was used.

b. Compares top 20% to bottom 20%.

c. An unspecified number of children repeated kindergarten and were sampled. Most children are likely younger than 6.

The impacts of *unplanned* pregnancies were studied in New Zealand [25] and, more recently, in the United Kingdom [31–33]. While the first study—again, with a smaller sample size—did not find meaningful differences, the Millennium Cohort study measured significant developmental differences using the Bracken and the British Ability Scales II in verbal and non-verbal measures, impacting both the socioemotional and literacy-numeracy domains.

In general, *mistimed* pregnancies followed the effect direction of unwanted or unplanned pregnancies reported in the preceding paragraphs but produced slightly weaker effects.

Similarly, *unintended* pregnancies—unwanted grouped with mistimed [27] or both mistimed and delayed pregnancies [28]—saw an attenuation of the effect.

## Discussion

### Main results

Our scoping review identified 8 "cohorts" with information on approximately 39,000 children born mostly in developed countries. The included studies examined the potential effects of delayed (n = 1), mistimed (n = 5), unintended (n = 2), unplanned (n = 2), and unwanted pregnancies (n = 7) on early child development within 53 years (1963–2016). While their results were not always consistent, overall, unwanted/unplanned pregnancies seemed to be associated with poorer child development when compared with wanted/planned pregnancies. Mistimed or delayed pregnancies correlated with weaker effects in the same direction. The Cochrane Handbook recommends against a meta-analysis given the differences in exposure and outcome measurement and selective reporting from the included studies [35].

### Limitations of the evidence

The existing evidence is limited to developed countries and Brazil (an upper-middle income economy), making it hard to translate their findings to populations in developing countries given varying levels of investment in social services and familial and socio-cultural contexts [36, 37]. For context, 92% of children are born in developing countries [38]. However, contexts are relevant as some of these countries were undergoing sociopolitical transformations when the studies were conducted. In the 1960s and 70s, Germany was navigating the path towards liberalizing abortion laws, which only became more permissive in 1976. Similarly, abortion was only partially decriminalized in Canada in 1969, and access to services remained inconsistent across provinces. The Christchurch study coincided with the nation's Contraception, Sterilisation, and Abortion Act in 1977, implying restricted access to abortion services at that time. Finally, while 'Roe v. Wade' significantly increased women's access to abortion in the US, starting in 1973 [39], disparities persisted among marginalized communities that also had historical reports of higher levels of unintended pregnancies (i.e., Hispanics and African Americans) [40].

The included studies were highly heterogeneous, reporting on different dimensions of development through scales that primarily relied on parental reports. Some reported the milestones at the time of assessment, and some assessed their child retrospectively, which is often less reliable [41]. Pregnancy intention/planning was assessed through different methods and at different periods (range: 0 to 5 years), which introduces the risk of recall bias [42]. Most studies assumed maternal age, marital status, household income, and child sex at birth acted as confounders. However, it is hard to determine how and to what extent differences in sampling and study design affected the precision and accuracy of the included estimands.

Our review did not find research on specific subgroups (e.g., females at birth, children who go through the adoption process after being born unwanted), except for one study that investigated maternal education [32]. This lack of differentiation between subgroups is a limitation of the existing literature because, in countries such as India, where there is a strong preference for male children, inequitable choices around healthcare access and nutrition lead to differential child health outcomes post-pregnancy [43]. While there is a significant overlap between adopted and unwanted/unplanned children, our review did not find any study that looked specifically at this group. Future studies could focus on studying the impact of unwanted pregnancy on child development in this group and whether adoption can help overcome the potential delays or inadequacies. Equity-based analyses were also missing in most of the studies.

## Limitations of the review process

Our search strategy used text-mining and keyword co-occurrence networks of reports identified *a priori*, which influenced both our terms and databases. With the increased accuracy of our search came a small number of results. Therefore, we might have missed studies published elsewhere, uncited by either of the reports that went through full-text screening. We also did not search the grey literature; publication bias might have skewed our results toward those studies that had statistically significant results. However, these studies often come from birth cohorts that address multiple outcomes. Given some logistics constraints, all studies underwent single screening, which introduces the risk of selection bias [44]. Some studies were conducted several decades ago and produced selective reporting. Even after contacting the authors, the original data was no longer accessible, and it was impossible to recalculate all potential outcomes.

Outcome reporting was also challenging. The included studies used different adjustment sets even within the same publication. While we did our best to unify all the results under the same criteria (specifically for the outcomes), the definition of the exposure and the adjustment variables changed significantly from study to study. Moreover, we did not analyze mediation of moderation results that tried to offer a more nuanced view of the relationship between pregnancy intention/planning and child development (i.e., categories of variables groups in nested models) because it was outside the scope of the project. It is imperative to recognize the nuances and complexities in the relationship between pregnancy intention/planning and child development, which is why we provided a narrative synthesis and a model compilation so that readers can more easily access the existing literature for a more in-depth understanding of the interplay between these factors at the population-level.

## Significance of the results

Children born from unintended or unplanned pregnancies seem to have poorer development during early childhood. A meta-analysis of over a million children showed that early intelligence during childhood is, in turn, associated with a 24% reduction in mortality during the 19–70 years follow-up period [45], which is why interventions that prevent cognitive delays had been prioritized as part of the Sustainable Development Goals [46].

Safeguarding and expanding reproductive rights are key aspects of improving health and social outcomes for women. Our results suggest that protecting these rights could have further positive impacts on other developmental goals. Comprehensive sexual education for both genders, along with unfettered access to a spectrum of contraception methods—including "morning-after" pills—and abortion services [47], are essential not only for women's autonomy but for the subsequent well-being and development of their children. Individuals empowered with knowledge and the means to control their reproductive choices are better positioned to provide for their children's needs, thereby influencing their educational opportunities and outcomes. Our review suggests that pre-conception strategies could contribute directly to the objective of ensuring stronger educational outcomes. Post-conception strategies are more challenging; pregnancy intention reports are not as reliable at the individual level as they are at the population level. The reasons behind unintended pregnancies, such as personal preferences, financial stability, emotional readiness, socioeconomic challenges, and broader social inequities within a specific context, can significantly impact early child development [48]. The effectiveness of strategies aimed at promoting early child development, including home-based approaches [49, 50], may vary based on these characteristics. Tailored support may need to be flexible and adaptive to address changing sentiments during and after pregnancy. Finally, women who have experienced multiple unintended pregnancies could also be targeted by

other policy efforts, given that other interventions might be less effective on them (e.g., women forced to conceive). Although some researchers advocate strengthening adoption referral systems when termination is not an option [47], children in institutional care often face increased risks of poor development and abuse when early adoption or other family-type care are unavailable [51]. Birth alerts, though intended as protective measures, have often had the unintended consequence of perpetuating systemic discrimination, particularly against Indigenous mothers in Canada [52, 53]. Such practices, especially when intersecting with issues like unintended pregnancies, can exacerbate the already profound social inequities and challenges these marginalized populations face [52, 53].

Our findings reinforce the notion that "child development starts at conception" [54]. Ensuring adequate development during early childhood, as mandated by target 2 on SDG 4 necessitates more than ECD investment. It requires supportive environments for women that respect and acknowledge the complexities of their reproductive choices and emotions; ensuring they have support and access to safe reproductive health options, including abortion, irrespective of their initial feelings about conception [48].

## Funding of the included studies

Stott & Latchford [26] were financed by the Department of National Health and Welfare (Canada), Ontario Ministry of Health, Ontario Ministry of Education. Dr. Petra Netter [24] received support from the German Research Foundation. Fergusson & Horwood [25] were financed by the Medical Research Council of New Zealand and NZ' National Children's Health Research Foundation. Three works received support from the National Institute of Child Health and Human Development (grant numbers: HD-30318, R01-HD35949 & R01-HD32330) [11, 28, 29]. The studies published by Carson and colleagues [31, 33] received support from the UK's Medical Research Council (grant G0701009), whereas the third study using the same dataset declared their work was based on the United Kingdom Millennium Cohort Study, which is funded by the United Kingdom Economic and Social Research Council and a consortium of government departments [32]. Two studies did not provide funding information or declarations about potential conflict of interest [30, 34]. One study declared no conflict of interest but did not provide specifics about funding [27].

## Supporting information

**S1 Appendix. Search strategy.**
(DOCX)

**S2 Appendix. List of excluded studies with reason.**
(DOCX)

**S3 Appendix. Preferred Reporting Items for Systematic reviews and Meta-Analyses extension for Scoping Reviews (PRISMA-ScR) checklist.**
(DOCX)

## Acknowledgments

The authors would like to acknowledge the contributions of Dr. Veena Sriram and Dr. Michael Law to earlier versions of this manuscript.

## Author Contributions

**Conceptualization:** Jorge Andrés Delgado-Ron, Magdalena Janus.

**Data curation:** Jorge Andrés Delgado-Ron.

**Formal analysis:** Jorge Andrés Delgado-Ron.

**Investigation:** Jorge Andrés Delgado-Ron, Magdalena Janus.

**Methodology:** Jorge Andrés Delgado-Ron, Magdalena Janus.

**Project administration:** Jorge Andrés Delgado-Ron.

**Resources:** Jorge Andrés Delgado-Ron.

**Software:** Jorge Andrés Delgado-Ron.

**Validation:** Magdalena Janus.

**Writing – original draft:** Jorge Andrés Delgado-Ron.

**Writing – review & editing:** Jorge Andrés Delgado-Ron, Magdalena Janus.

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
