## [Decision Letter · Decision Letter 0]

16 Aug 2023

PGPH-D-23-00602

Association between pregnancy planning or intention and early child development: a systematic scoping review

Dear Dr. Delgado-Ron,

Thank you for submitting your manuscript to PLOS Global Public Health. After careful consideration, we feel that it has merit but does not fully meet PLOS Global Public Health’s publication criteria as it currently stands. Therefore, we invite you to submit a revised version of the manuscript that addresses the points raised during the review process.

Please consider the specific comments of the reviewer especially with respect to the introduction and discussion and how pregnancy conception is conceptualized. Provide an itemized response detailing your changes for each of their concerns with any resubmission.

We look forward to receiving your revised manuscript.

Kind regards,

Colleen M. Davison

Academic Editor

Journal Requirements:

Additional Editor Comments (if provided):

Reviewers' comments:

Reviewer's Responses to Questions

**Comments to the Author**

1. Does this manuscript meet PLOS Global Public Health’s publication criteria? Is the manuscript technically sound, and do the data support the conclusions? The manuscript must describe methodologically and ethically rigorous research with conclusions that are appropriately drawn based on the data presented.

Reviewer #1: Partly

2. Has the statistical analysis been performed appropriately and rigorously?

Reviewer #1: I don't know

3. Have the authors made all data underlying the findings in their manuscript fully available (please refer to the Data Availability Statement at the start of the manuscript PDF file)?

Reviewer #1: Yes

4. Is the manuscript presented in an intelligible fashion and written in standard English?

Reviewer #1: Yes

5. Review Comments to the Author

Reviewer #1: This systematic scoping review presents a unique approach to considering reproductive rights from the perspective of pregnancy intention and impacts on child development. The review outlines a comprehensive search and detailed strategy.

Overall, there is a lack of discussion regarding the interpretation of pregnancy intention, as well as engagement with the sociopolitical contexts and nuanced experiences and perspectives of women who have unwanted/unintended/mistimed pregnancies. This relates to concerns around the discussion in the Significance of Results section and interpretation of the review results as outlined below.

Lines 330-334 under Significance of results seem to be implying that poorer early child development stems from women/families/circumstances in which children are “not loved enough”. This seems like a large assumption that overlooks many different contexts and experiences of women that seems problematic for several reasons:

1. It assumes that intention is a constant across the pregnancy and child life course – once unwanted, always unwanted. Not only could there be errors or misinterpretations regarding the meaning behind intention for both researcher and participant but intention could also be a dynamic process – perspectives regarding pregnancy/parenting could shift over time, throughout the course of the pregnancy etc. See Santelli et al., 2003 ; Santelli et al., 2009; Borrero et al., 2014 for more on this.

2. It assumes pregnancy intention and love are the same or directly related. Women may not have wanted or planned for these pregnancies because they knew they were unable to provide for the child due to a variety of circumstances related to capacity and social inequities rather than lack of love. In fact, arguably not wanting a pregnancy could be an expression of love – love for oneself, love for one’s existing family, even love for the idea of an unborn child if they believe carrying and birthing that child would deliver them into harm/oppression etc. See Manze et al., 2021; Leath et al., 2022 as examples related to this.

3. Attributing poorer early child development to “not being loved enough” overlooks the context – that women may not want their pregnancy at the time for a whole range of different reasons, and are situated within a range of different social/physical environments, that could impact the child’s development (some of which are briefly touched on in confounders but certainly not all). The reasons regarding pregnancy intention or not wanting a pregnancy could range from personal preference, personal capacity, social environment, inequities etc. and that these reasons all function differently in terms of pregnancy and parenting behaviours. See Auerbach et al., 2023 as an example related to this.

Although the authors looked at studies from six countries: New Zealand (NZ), Germany, Canada, Brazil, the United States, and the United Kingdom, they don’t contextualize the information within various sociopolitical contexts. This has potentially important implications because the study is not just talking about women who may have wanted an abortion but also women who are parenting children born of those unintended/mistimed pregnancies:

1. There are certain populations within these countries that may be more affected by lack of access to birth control, abortion services etc. and more likely to have higher rates of unwanted/unintended/mistimed pregnancies such as underserved and racialized populations in the United States and Canada. Given that it is largely structurally marginalized women that have difficulty accessing abortion services/reproductive services within these countries etc., a focus on “love” in the discussion may potentially reinforce stereotypes of “bad motherhood” related to marginalized women who may not have planned for their pregnancies but are parenting those children now.

2. The proposed interventions/actions in lines 342-347 are unclear in terms of specifics and seem to focus on children born into families of unintended pregnancies. Given that certain populations are more likely to have unwanted/unintended/mistimed pregnancies AND are more likely to face social inequities, racism/discrimination etc. within these countries it would be helpful to provide more details as the recommendations potentially targets the women and children in a way that also seems to ignore the dynamic nature of pregnancy/parenting intention and the socio-political context. For instance, flagging women who have unintended pregnancies as requiring intervention could lead to harmful practices, especially for marginalized women, who face discrimination, birth alert practices, inequitable child welfare practices etc.

While I agree with the authors that there is a potential need to create supportive environments where women who do not desire pregnancy can access abortion, and that conception should be a desired life event, and can understand the limitations around discussing many complex concepts, overall the discussion and interpretation of results seems to both erase women (their knowledge/experiences/positions) in their lives AND yet almost villainizes them within the context of child development. Re-writing the introduction and discussion sections to better contextualize women’s experiences and perspectives and considerations of the interpretation of intention would help to address some of the concerns related to the interpretation and significance of the systematized review results.

Additional considerations:

1. Lines 339-342 reference a goal of lifelong learning. It’s unclear what goal/objective this sentence is referring to – is it presumed the woman’s goal of lifelong learning?

2. Lines 324-326 state the authors overlooked other results that tried to offer a more nuanced view of the relationship. It’s unclear the rationale behind this – was it for ease of analysis? How does this relate overall to the interpretation and understanding of the results?

6. PLOS authors have the option to publish the peer review history of their article (what does this mean?). If published, this will include your full peer review and any attached files.

**Do you want your identity to be public for this peer review?** For information about this choice, including consent withdrawal, please see our Privacy Policy.

Reviewer #1: No

---

## [Decision Letter · Decision Letter 1]

1 Nov 2023

Association between pregnancy planning or intention and early child development: a systematic scoping review

PGPH-D-23-00602R1

Dear Dr. Delgado-Ron,

We are pleased to inform you that your manuscript 'Association between pregnancy planning or intention and early child development: a systematic scoping review' has been provisionally accepted for publication in PLOS Global Public Health.

Before your manuscript can be formally accepted you will need to complete some formatting changes, which you will receive in a follow up email. A member of our team will be in touch with a set of requests. Please also see the small edit required by reviewer #1 on page 276.

Best regards,

Colleen M. Davison

Academic Editor

Reviewer Comments (if any, and for reference):

Reviewer's Responses to Questions

**Comments to the Author**

1. If the authors have adequately addressed your comments raised in a previous round of review and you feel that this manuscript is now acceptable for publication, you may indicate that here to bypass the “Comments to the Author” section, enter your conflict of interest statement in the “Confidential to Editor” section, and submit your "Accept" recommendation.

Reviewer #1: All comments have been addressed

2. Does this manuscript meet PLOS Global Public Health’s publication criteria? Is the manuscript technically sound, and do the data support the conclusions? The manuscript must describe methodologically and ethically rigorous research with conclusions that are appropriately drawn based on the data presented.

Reviewer #1: Yes

3. Has the statistical analysis been performed appropriately and rigorously?

Reviewer #1: I don't know

4. Have the authors made all data underlying the findings in their manuscript fully available (please refer to the Data Availability Statement at the start of the manuscript PDF file)?

Reviewer #1: Yes

5. Is the manuscript presented in an intelligible fashion and written in standard English?

Reviewer #1: Yes

6. Review Comments to the Author

Reviewer #1: The authors have thoughtfully considered and addressed all previous reviewer comments. The authors have made a considerable effort to more explicitly reference the sociopolitical contexts and effects of social inequities and structural marginalization regarding pregnancy intention and outcomes.

My only additional edit is regarding line 276 where the sentence reads "However, lessons are relevant" - perhaps I am unfamiliar with the terminology but "lessons" seems to be referring to "contexts", as it is unclear what other "lessons" this could mean. Perhaps, consider changing the term for clarity purposes.

Overall, this manuscript represents a unique approach to reproductive rights research and adds to the discussion regarding child development and pregnancy intention.

7. PLOS authors have the option to publish the peer review history of their article (what does this mean?). If published, this will include your full peer review and any attached files.

**Do you want your identity to be public for this peer review?** For information about this choice, including consent withdrawal, please see our Privacy Policy.

Reviewer #1: No
